# Circuits that encode and guide alcohol-associated preference

**Kristin M Scaplen[1], Mustafa Talay[1†], Kavin M Nunez[2], Sarah Salamon[3], Amanda G Waterman[1], Sydney Gang[4], Sophia L Song[1], Gilad Barnea[1], Karla R Kaun[1]\***

[1]Department of Neuroscience, Brown University, Providence, United States; [2]Department of Molecular Pharmacology and Physiology, Brown University, Providence, United States; [3]Department of Pharmacology, University of Cologne, Cologne, Germany; [4]Department of Biochemistry, Brown University, Providence, United States

**Abstract** A powerful feature of adaptive memory is its inherent flexibility. Alcohol and other addictive substances can remold neural circuits important for memory to reduce this flexibility. However, the mechanism through which pertinent circuits are selected and shaped remains unclear. We show that circuits required for alcohol-associated preference shift from population level dopaminergic activation to select dopamine neurons that predict behavioral choice in *Drosophila melanogaster*. During memory expression, subsets of dopamine neurons directly and indirectly modulate the activity of interconnected glutamatergic and cholinergic mushroom body output neurons (MBON). Transsynaptic tracing of neurons important for memory expression revealed a convergent center of memory consolidation within the mushroom body (MB) implicated in arousal, and a structure outside the MB implicated in integration of naïve and learned responses. These findings provide a circuit framework through which dopamine neuronal activation shifts from reward delivery to cue onset, and provide insight into the maladaptive nature of memory.

**\*For correspondence:**
karla_kaun@brown.edu

**Present address:** [†]Howard Hughes Medical Institute, Department of Molecular and Cellular Biology, Harvard University, Cambridge, United States

**Competing interests:** The authors declare that no competing interests exist.

## Introduction

An organism's behavior is guided by memories of past experiences and their associated positive or negative outcomes. Long-term memory retention requires the strengthening of labile memory traces so they are available for future retrieval. However, successful associations are also dynamic and malleable providing opportunities for updating associations based on new information. Thus, in order for organisms to adapt to their environment, they must find a balance between the persistence and flexibility of memories (*Richards and Frankland, 2017*).

In substance use disorder (SUD), the balance between memory persistence and flexibility is often absent or difficult to achieve (*Font and Cunningham, 2012*; *Torregrossa and Taylor, 2013*; *Hitchcock et al., 2015*; *American Psychiatric Assocation, 2013*). Alcohol similarly disrupts memory systems resulting in enduring preferences, attentional bias for associated cues, and habitual behaviors (*Fadardi et al., 2016*; *Field and Cox, 2008*; *Everitt and Robbins, 2005*; *Corbit et al., 2012*; *Gerdeman et al., 2003*; *Yin, 2008*; *Hyman et al., 2006*; *Robinson and Berridge, 2003*; *Goodman and Packard, 2016*; *White, 1996*). In alcohol use disorder (AUD), preference and cravings for alcohol persist in the face of aversive consequences, leading to maladaptive drug seeking behaviors and ultimately a devastating economic and social impact on individuals, communities, and society as a whole (*WHO, 2018*). Understanding the circuitry mechanisms that underlie the encoding and expression of alcohol-associated memories is critical to understanding why these memories are resistant to change.

A significant effort has been devoted to identifying and investigating circuitry changes as a consequence of alcohol (*Lovinger and Alvarez, 2017*; *Corbit and Janak, 2016*; *Corbit et al., 2012*; *Keiflin and Janak, 2015*; *Dong et al., 2017*; *Stuber et al., 2010*; *Volkow and Morales, 2015*; *Volkow et al., 2013*). The neuronal, genetic, and physiologic diversity that exists within the mammalian brain, however, has made this task challenging (*Morales and Margolis, 2017*). *Drosophila melanogaster* is a powerful model organism to address these challenges because of its lower complexity and the availability of neurogenetic tools that permit dissection of memory circuits with exact temporal and spatial resolution. Further, the neural circuits underlying the *Drosophila* reward response are remarkably similar to mammals (*Scaplen and Kaun, 2016*). *Drosophila* form persistent appetitive memories for the pharmacological properties of alcohol that last up to 7 days post acquisition and impel flies to walk over a 120V electric shock in the presence of associated cues (*Kaun et al., 2011*; *Nunez et al., 2018*). This suggests that *Drosophila* and mammalian alcohol-associated memories are similarly inflexible in the face of aversive consequences.

We sought to identify the circuits important for alcohol-associated memories using a multipronged approach combining behavioral, thermogenetic, in vivo calcium imaging, and transsynaptic tracing. We show that circuits required for formation of alcohol preference shift from population-level dopaminergic encoding to two microcircuits comprising of interconnected dopaminergic, glutamatergic, and cholinergic neurons. Circuits required for the expression of alcohol-associated memories converge onto a mushroom body output neuron (MBON) that regulates consolidation and the fan-shaped body (FSB), a higher-order brain center implicated in arousal and modulating behavioral response (*Donlea et al., 2018*; *Pimentel et al., 2016*; *Troup et al., 2018*; *Qian et al., 2017*; *Weir and Dickinson, 2015*; *Weir et al., 2014*; *Hu et al., 2018*; *Liu et al., 2006*). Our results provide an in vivo circuit framework for how drugs of abuse temporally regulate acquisition and expression of sensory memories, which ultimately results in a shift in behavioral response from malleable to inflexible.

## Results

### Dopamine neurons innervating the mushroom body are required for alcohol reward associations

Dopamine has a long-standing role in addiction and a defined role in reward-related behavioral learning that spans across species (*Wanat et al., 2009*; *Yoshimoto et al., 1992*; *Hyman et al., 2006*; *Robbins and Everitt, 2002*; *Torregrossa et al., 2011*; *Kaun et al., 2011*; *Scaplen and Kaun, 2016*). In *Drosophila,* the establishment of alcohol-associated preference requires a central brain structure called the mushroom body (MB) and dopamine neurons (DANs) (*Kaun et al., 2011*). It is unclear, however, which population of DANs are necessary for alcohol-associated preference. A discrete population of protocerebral anterior medial (PAM) DANs that innervate the MB have an identified role in detecting and processing natural rewards (*Liu et al., 2012*; *Yamagata et al., 2015*; *Huetteroth et al., 2015*; *Lin et al., 2014*). PAM neurons are required for the acquisition of sucrose and water reward memories, are activated by sucrose and water administration (*Harris et al., 2015*; *Liu et al., 2012*; *Lin et al., 2014*), and artificial activation is sufficient to induce reward memories (*Burke et al., 2012*; *Yamagata et al., 2015*). Thus, we first tested whether PAM neurons were also required for alcohol-associated preference (*Figure 1A*).

For selective manipulations of PAM neurons, we expressed the dominant negative temperature sensitive *shibire* (*shi^{ts}*) using *R58E02-GAL4* (*Liu et al., 2012*). To establish temporal requirements, we temporarily and reversibly inactivated neurotransmission by raising the temperature to restricted levels (30˚C) during memory acquisition, the overnight consolidation period, or memory retrieval. Acquisition was defined as the time during which an odor was presented in isolation (unpaired odor) for 10 min followed by a second odor that was paired with an intoxicating dose of vaporized ethanol (paired odor + ethanol) for an additional 10 min. During acquisition, reciprocally trained flies received three of these spaced training sessions. Post-acquisition, flies were given a choice between the odor that was previously presented with an intoxicating dose of ethanol and the odor that was presented in isolation (*Figure 1A*). Retrieval was measured in a Y-maze 24 hr post acquisition and defined as the time during which the flies chose between the previously presented odors. Inactivating neurotransmission in PAM DANs during acquisition or retrieval, but not during the overnight

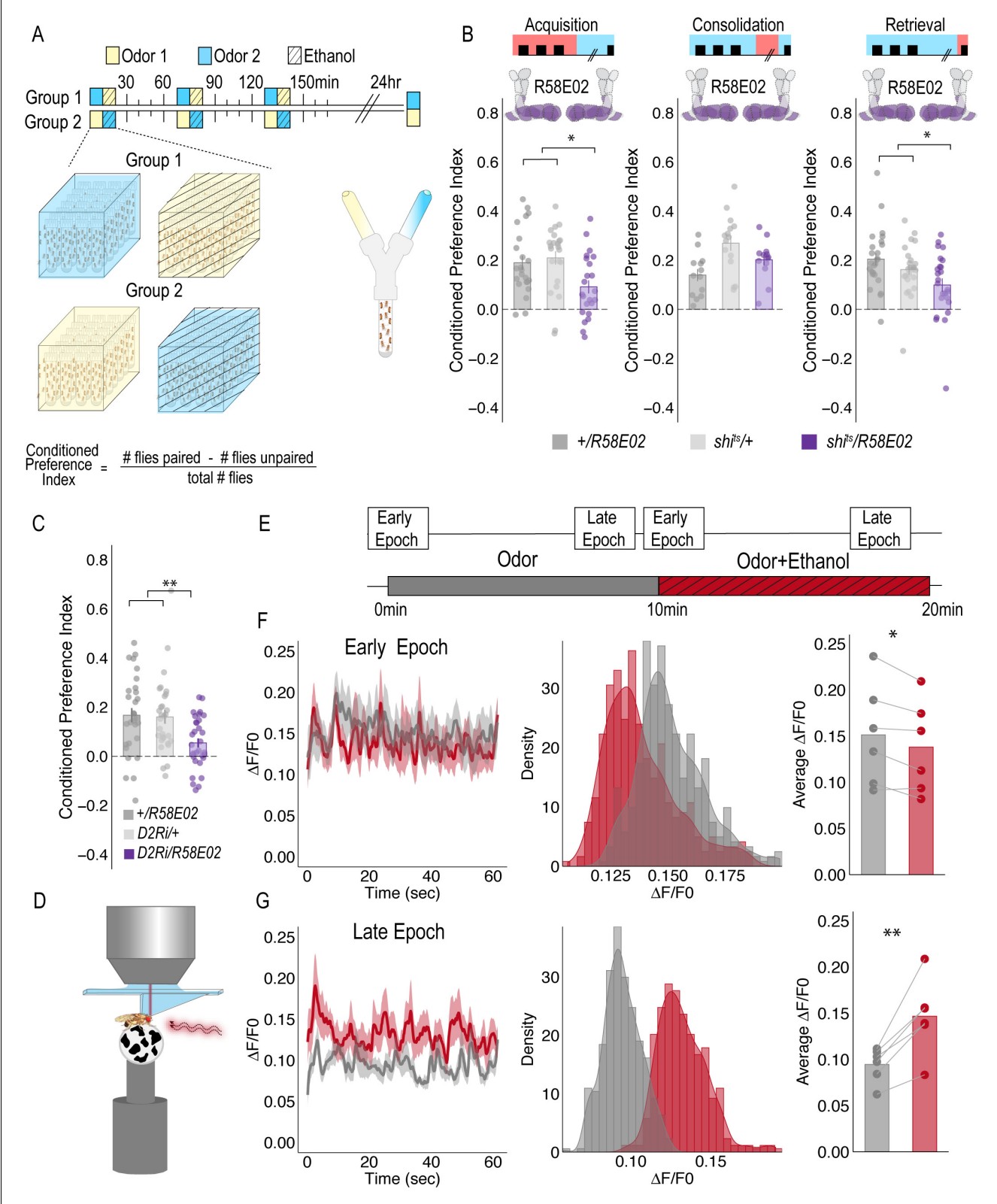

**Figure 1.** PAM DANs are necessary for encoding alcohol-associated preference. (**A**) Schematic illustrating odor condition preference paradigm. Vials of 30 flies are presented with three sessions of 10 min of an unpaired odor, followed by 10 min of a paired odor plus intoxicating vaporized ethanol. To control for odor identity, reciprocal controls were used. Flies were tested 24 hr later in a standard Y maze (**B**) PAM dopaminergic neurons activity is necessary during acquisition (F(2, 66)=5.355, p=0.007) and retrieval (F(2,71)=5.707, p=0.005), but not consolidation. Bar graphs illustrate mean +/-

*Figure 1 continued on next page*

*Figure 1 continued*

standard error of the mean. Raw data are overlaid on bar graphs. Each dot is an n of 1, which equals approximately 60 flies (30 per odor pairing). One-way ANOVA with Tukey Posthoc was used to compare mean and variance. *p<0.05 (C) RNAi knockdown of D2R within the PAM population targeted using the R58E02 GAL4 driver significantly reduced alcohol-associated preference F(2,89)=6.441, p=0.002. (D) Schematic illustrating calcium imaging paradigm. (E) Flies are exposed to odor followed by odor plus intoxicating vaporized ethanol while resting or walking on a ball. We used the same odor for both conditions so we could better compare circuit dynamics in response to ethanol and control for odor identity. Fluorescence was captured for 61 s recording epochs that were equally spaced by 2 min. (F). Average traces recorded during early odor and odor plus ethanol exposures. Middle panels illustrate the binned ΔF/F0 and highlights a change in calcium dynamics as a consequence of ethanol exposure. Right panels illustrate the average ΔF/F0 for each fly in each condition. Early Epochs of odor plus ethanol had significantly lower signal (F(1,5)=8.705, p=0.03). (G) Average traces recorded during late odor and odor plus ethanol exposures. Middle panels illustrate the binned ΔF/F0 and highlights a change in calcium dynamics as a consequence of ethanol exposure. Right panels illustrate the average ΔF/F0 for each fly in each condition. Late Epochs of odor plus ethanol had significantly higher signal (F(1,5)=24.177, p=0.004). Within Subject Repeated Measures ANOVA was used to compare mean and variance across condition and time. Scale bar = 50 μm *p<0.05 **p<0.01.

The online version of this article includes the following figure supplement(s) for figure 1:

**Figure supplement 1.** Although inactivation of PAM neurons increased group flies in an open field arena (n = 15), it did not affect alcohol induced activity suggesting that a decrease in preference is encoded independently from the amount of activity animals exhibit while intoxicated (*Figure 1G*).

**Figure supplement 2.** Dopamine staining within the brain following 10 min of air or 10 min of ethanol.

**Figure supplement 3.** Calcium Imaging from terminals of PAM population of DANs in response to odors and ethanol.

**Figure supplement 4.** Requirement of PAM DANs and Kenyon cells in formation of alcohol-associated preference.

**Figure supplement 5.** Subsets of PAM DANs are dispensable for encoding alcohol-associated preference.

**Figure supplement 6.** Subsets of PAM DANs are required for retrieval, but not acquisition or consolidation.

**Figure supplement 7.** mRNA quantification of dopamine receptors (DRs) in all neurons following constitutive expression of DR-RNAi's.

**Figure supplement 8.** Temperature controls for DAN inhibition experiments that showed decreases in retrieval of alcohol associated preference at the restrictive temperature.

consolidation, significantly reduced preference for cues associated with ethanol (*Figure 1B*). Further decreasing dopamine-2-like receptors (D2R), which are thought to act as auto-receptors, (*Vickrey and Venton, 2011*), in PAM neurons significantly reduced preference for cues associated with ethanol suggesting that the regulation of dopamine release at the synapse is important for alcohol reward memory (*Figure 1C*).

Strikingly, despite dopamine's established role in modulating locomotor and motor responses (*da Silva et al., 2018*; *Howe and Dombeck, 2016*; *Dodson et al., 2016*; *Syed et al., 2016*; *Lima and Miesenböck, 2005*; *Romo and Schultz, 1990*; *Schultz, 2007*), inactivating all PAM dopaminergic neurons did not disrupt ethanol induced activity (*Figure 1—figure supplement 1*). Together, these results demonstrate that PAM neurons are required for encoding preference, but not for the locomotor response to the acute stimulatory properties of ethanol, and dopamine regulation at the synapse is important for memory.

## Dopaminergic encoding of alcohol memory acquisition occurs at the population level

To determine how alcohol influenced activity of PAM DANs, we first used a dopamine staining protocol to label dopamine within the brain following 10 min of air or alcohol. As expected, there was a significant amount of dopamine labeled within the mushroom body and the majority of fluorescence was limited to the horizontal lobes (*Figure 1—figure supplement 2*). We hypothesized that dopamine fluorescence would increase within the horizontal lobes of the MB in response to alcohol. Quantification of fluorescence revealed a trending increase in dopamine that was not statistically different from control (*Figure 1—figure supplement 2*). We reasoned that dopamine staining likely could not distinguish between dopamine in the presynaptic terminals and dopamine in the synaptic cleft. Thus, we turned to 2-photon functional calcium imaging to monitor circuitry dynamics of PAM dopaminergic activity in the context of intoxicating alcohol.

We used *R58E02-Gal4* to express *GCaMP6m* (*Chen et al., 2013*) and recorded from the PAM presynaptic terminals at the MB while naïve flies were presented with 10 min of odor, followed by 10 min of odor plus intoxicating doses of alcohol (*Figure 1C*). Interestingly, early in the respective recording sessions (odor vs odor + alcohol), changes in calcium dynamics was greater in the odor only group (*Figure 1D*), however with prolonged alcohol exposure, greater calcium dynamics started to emerge in the odor + alcohol group (*Figure 1E*). Similar effects were not evident if the fly was

presented with two different odors alone or alcohol alone (*Figure 1—figure supplement 2*), suggesting that the reported effects are not merely a consequence of odor identity or the pharmacological properties of alcohol, but perhaps unique to alcohol associations.

To address whether specific subsets of dopamine neurons within the PAM neuron population are necessary for alcohol-associated preference, we blocked transmission in subsets of these neurons using 18 highly specific split-Gal4 lines during both acquisition and retrieval. We found that preference was disrupted when neurotransmission was blocked in DANs projecting to the medial aspect of horizontal MB (*Figure 1—figure supplement 4A*). Similar disruptions were evident when neurotransmission was blocked in intrinsic MB Kenyon cells (*Figure 1—figure supplement 4B*). We therefore selected split-Gal4 lines that targeted the medial aspect of the horizontal lobe and determined their role specifically in acquisition of alcohol-associated preference. Surprisingly, unlike 24 hr sucrose memory (*Ichinose et al., 2015*; *Yamagata et al., 2015*; *Huetteroth et al., 2015*), thermogenetic inactivation of specific subsets of DANs, innervating compartments of the medial horizontal lobe during acquisition did not disrupt 24 hr alcohol-associated preference (*Figure 1—figure supplement 5*). It's unlikely this is a result of the lines being split-Gal4 lines since HL9, a PAM-specific *Dopamine decarboxylase (Ddc)* promotor-driven Gal4 (*Claridge-Chang et al., 2009*) shows a similar effect (*Figure 1—figure supplement 6*). Cell counts of the broadest split-GAL4 lines (40B and 42B), HL9, and R58E02 driver lines revealed that despite targeting nearly all of the horizontal lobes of the MB, 40B, 42B, and HL9 expressed in significantly fewer cells, and thus do not engage the entire population of PAM DANs (*Supplementary file 1—Table 1*). Together these data suggest that alcohol reward memories are encoded via a population of DANs involved in reward memory that progressively increase their activity as the flies become intoxicated.

## Memory expression is dependent on a sparse subset of dopamine neurons

A hallmark of reward-encoding DANs is the gradual transfer in response from reward delivery during learning to the cue that predicts a reward during expression of the associated memory (*Keiflin and Janak, 2015*; *Schultz, 2016*; *Schultz, 2015*). However, the circuit mechanisms underlying this shift and knowledge about whether all DANs respond to the predictive cue, or a selective subset of DANs is unknown. We temporally inactivated neurotransmission in subsets of DANs during retrieval to determine which subsets are required for a behavioral response to the predictive cue. Strikingly, only inactivating DANs innervating β′2a compartment of the MB, using split-Gal4 line *MB109B*, significantly reduced alcohol-associated preference, demonstrating that these neurons are important for the expression of alcohol-associated preference during retrieval (*Figure 2F*).

## A dopamine-glutamate circuit regulates memory expression

Our next goal was to map the circuits through which β′2a DANs drive behavioral choice. We tested the requirement of MB output neurons (MBONs) that align with β′2a DANs. Inactivating glutamatergic MBONs innervating similar compartments during acquisition using five different split-Gal4 lines, did not significantly reduce alcohol-associated preference (*Figure 3A–E*). However, similar inactivation during retrieval identified a single β2 β′2a glutamatergic MBON important for the expression of alcohol-associated preference (*Figure 3I*) thereby defining a putative retrieval microcircuit that consists of a subset of 8–10 dopamine neurons innervating the β′2a MB compartment and a single glutamatergic MBON that also innervates the β′2a MB compartment (β2 β′2a; *Figure 3L*).

Previous work suggested that β′2a DANs were anatomically connected with β′2amp MBONs at the level of the MB, however, it was unclear to which MBON β′2a DANs were synaptically connected (*Lewis et al., 2015*). To test connectivity between β′2a DANs and β2β′2a MBONs we used the recently developed anterograde transsynaptic labeling method *trans*-Tango to label the postsynaptic targets of the β′2a DANs (*Talay et al., 2017*; *Figure 4A*). Crossing split-Gal4 line MB109B with *trans*-Tango flies revealed α′β′ MB neurons as postsynaptic (*Figure 4Bi*). Interestingly, β′2mp MBON, and not β2 β′2a, MBON were labeled as post synaptic to β′2a DANs (*Figure 4Bii*).

It's possible that synaptic connectivity between β′2a DANs and β2β′2a MBON is not sufficient to be picked up by *trans*-Tango tools. We thus tested functional connectivity between DANs and β2 β′2a using dopamine receptor RNAi (*Figure 4—figure supplement 1A and B*). Reducing expression of D1-like receptors (D1Rs) or D2Rs in β2β′2a MBON did not disrupt alcohol-associated preference.

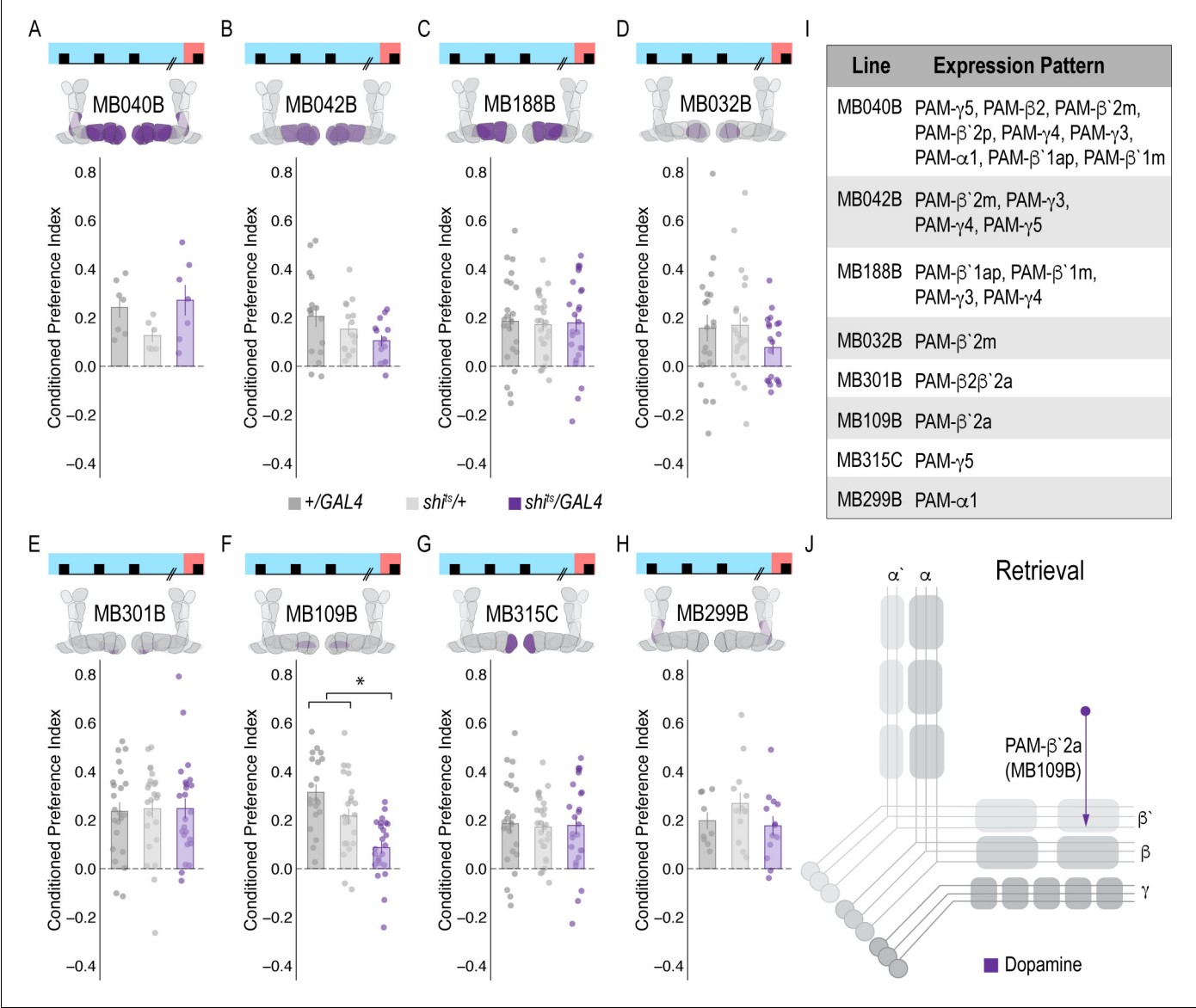

**Figure 2.** Memory expression during retrieval is dependent on a sparse population of DANs. (A–H) A thermogenetic approach was used to inactivate neurotransmission during retrieval, but not acquisition, in PAM DANs with varying expression patterns. (F) Inactivating β'2a DANs during retrieval significantly reduced preference for alcohol-associated cues. One-way ANOVA with Tukey Posthoc was used to compare mean and variance. $F_{(2,65)}$ =14.18, p=7.78×10−6. Bar graphs illustrate mean +/- standard error of the mean. Raw data are overlaid on bar graphs. Each dot is an n of 1, which equals approximately 60 flies (30 per odor pairing). (I) Chart illustrating the expression pattern of each split-GAL4 tested with intensity ranges of 2–5 (*Aso et al., 2014a*). (J) Model of circuits responsible for expression of alcohol-associated preference during retrieval, which highlights the importance of sparse subsets of dopaminergic activity during retrieval for the expression of alcohol-associated preference. *p<0.01.

Previous work from our lab reported the requirement of D2Rs in intrinsic MB neurons for alcohol-associated preference (*Petruccelli et al., 2018*), suggesting an indirect D2R pathway that regulates expression of alcohol memory.

## A separate dopamine-glutamate circuit regulates memory consolidation

Transsynaptic tracing revealed a putative direct synaptic connection between β'2a DANs and β'2mp glutamatergic MBONs in regulating alcohol-associated preference (*Figure 4Bii*). We tested whether this connection was functionally important in regulating alcohol-associated preference using dopamine receptor RNAi lines. Decreasing levels of D2R, but not D1Rs, reduced alcohol-associated

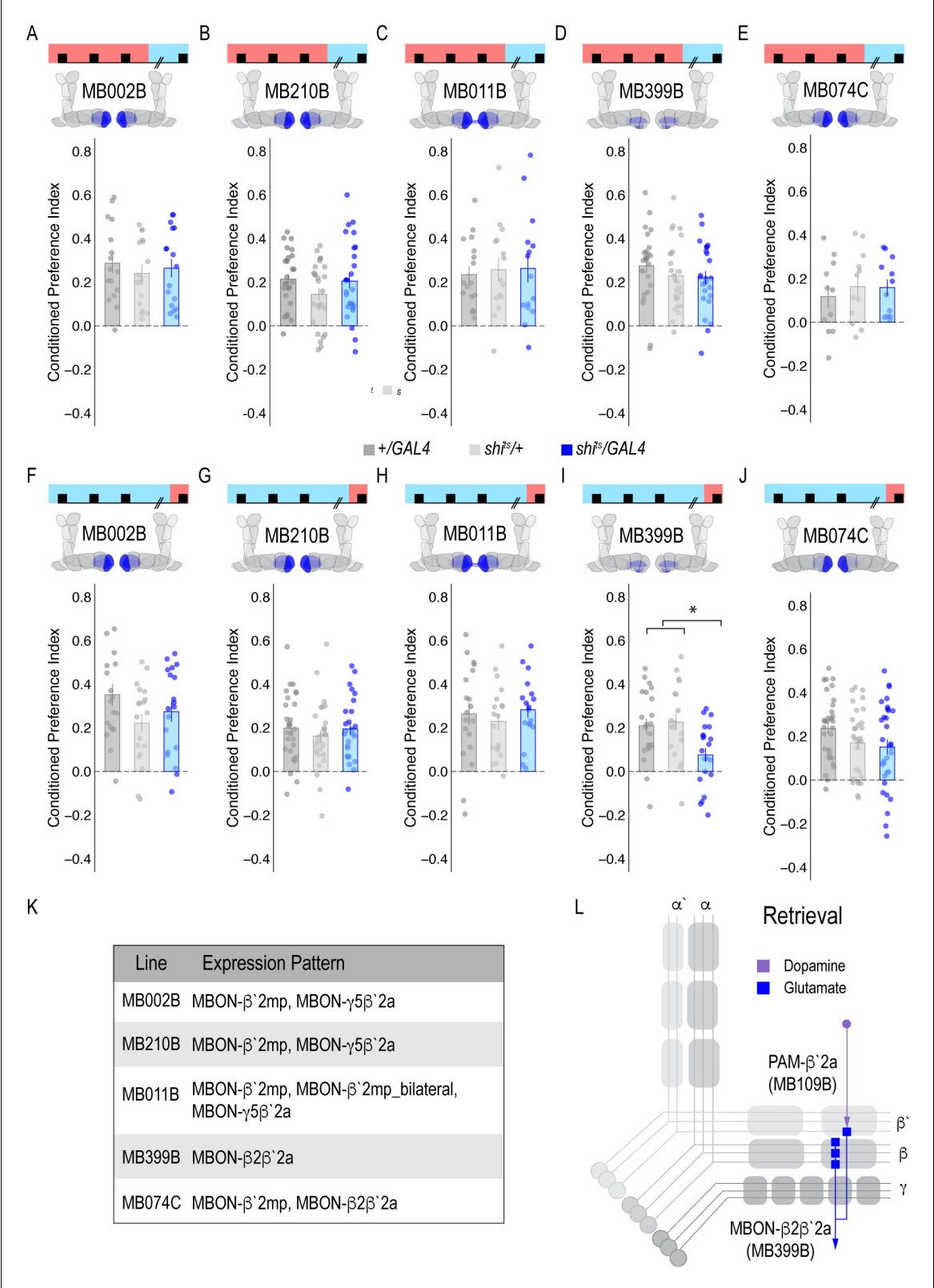

**Figure 3.** Memory expression during retrieval, but not acquisition, is dependent on a sparse population of glutamatergic MBONs. (A–E) Thermogenetic inactivation of glutamatergic MBONs innervating similar compartments to β'2a PAM DANs during acquisition did not disrupt encoding of alcohol-associated preference. (F–J) However, inactivating neurotransmission during retrieval revealed the specific importance of MBON β2β'2a glutamatergic neuron (I) for the expression of alcohol-associated preference F(2,59)=5.62, p=0.006. One-way ANOVA with Tukey Posthoc was used to compare mean

*Figure 3 continued on next page*

and variance. *p<0.01 Bar graphs illustrate mean +/- standard error of the mean. Raw data are overlaid on bar graphs. Each dot is an n of 1, which equals approximately 60 flies (30 per odor pairing). (**K**) Chart illustrating the expression pattern of each split-GAL4 tested with intensity ranges of 2–5 (*Aso et al., 2014a*). (**L**) Updated model of circuits responsible for expression of alcohol-associated preference. Retrieval circuits require specific subsets of DANs and a single MBON glutamatergic neuron innervating the β2'a compartment.

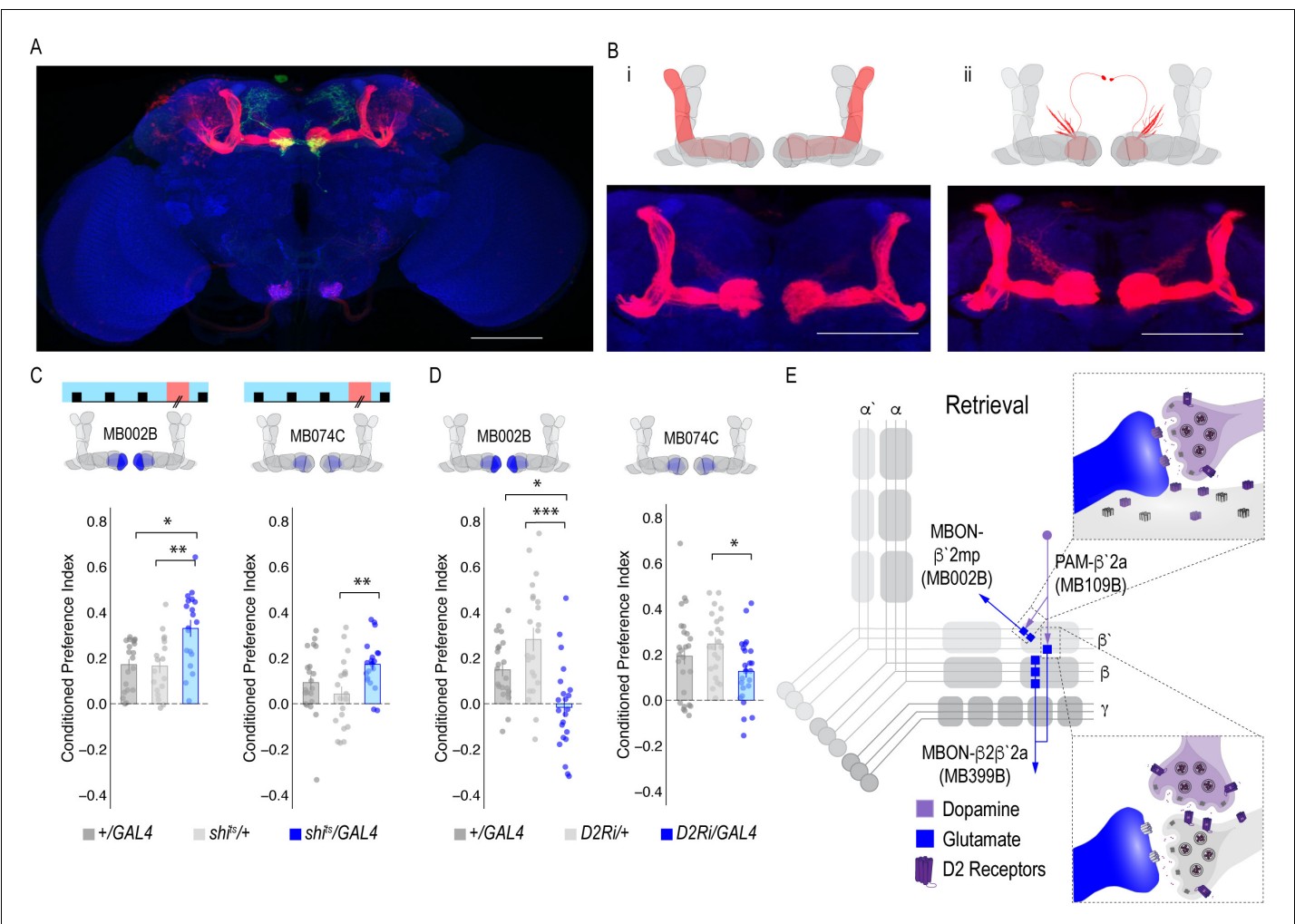

**Figure 4.** MBON β'2mp glutamatergic neuron is postsynaptic to β'2a PAM DANs and important for memory consolidation. (**A**) Representative maximum projection confocal stacks of MB109B > *trans*Tango. (**B**) *trans*-Tango reveal the α'/β' MB lobe (**i**) and MBON β'2mp neurons (**ii**) as postsynaptic to β'2a DANs. (**C**) Thermogenetic inactivation of MBON β'2mp during consolidation using MB002B significantly increased alcohol reward preference $F_{(2,54)}$ = 9.287, p=0.0003. Thermogenetic inactivation of β'2mp during consolidation using MB074C significantly increased alcohol reward preference relative to UAS controls $F_{(2,71)}$ = 3.51, p=0.04. (**D**) Knockdown of D2R in MBON β'2mp using MB002B significantly decreased alcohol-associated preference $F_{(2,63)}$=12.77, p=2.22×10−05. Knockdown of D2R in MBON β'2mp using MB074C significantly decreased alcohol-associated preference relative to GAL4 controls $F_{(2,71)}$=3.51, p=0.04. One-way ANOVA with Tukey Posthoc was used to compare mean and variance. Bar graphs illustrate mean +/- standard error of the mean. *p<0.05 **p<0.01 (**f**) Circuits responsible for encoding alcohol-associated preference during retrieval. Scale bar = 50 μm.

The online version of this article includes the following figure supplement(s) for figure 4:

**Figure supplement 1.** RNAi knockdown of dopamine receptors in β'2 MBONs in alcohol-associated preference.

**Figure supplement 2.** Temperature controls for lines that showed decreases in retrieval of alcohol-associated preference at a restrictive temperature.

preference (*Figure 4D*, *Figure 4—figure supplement 1C*), providing functional evidence for a direct D2R-dependent pathway that regulates alcohol memory.

Previous work in *Drosophila* reported that activating β'2mp MBON promotes arousal (*Sitaraman et al., 2015*). Thus, we hypothesized that inactivating β'2mp MBON while flies normally sleep would further decrease arousal and facilitate memory consolidation. To test this hypothesis, we inactivated neurotransmission of β'2mp MBON using two different split-GAL4 driver lines (MB074C and MB002B) during the overnight consolidation period (*Aso et al., 2014a*). Despite having no effect during acquisition or retrieval (*Figure 3A,E,F,J*), inactivating the β'2mp MBON during overnight consolidation period enhanced alcohol-associated preference (*Figure 4C*). Together these data suggest that β'2a DANs inhibit the β'2mp glutamatergic MBON via D2R receptors which leads to the expression of alcohol-associated preference. In the absence of dopamine (*Figure 2F*) or D2R receptors (*Figure 4D*), preference is disrupted.

## Convergent microcircuits encode alcohol reward expression

The central role for the β'2mp MBON in consolidation suggests that this region may integrate information from several circuits required for memory expression. Previous anatomical studies predicted that β'2mp glutamatergic MBON and α'two cholinergic MBON were synaptically connected (*Aso et al., 2014*). *trans*-Tango experiments demonstrate that β'2mp MBON is indeed a postsynaptic target of the α'2 MBON (*Figure 5A*). We previously showed that inactivating the α'2 cholinergic MBON throughout both memory acquisition and expression decreased alcohol-associated preference (*Aso et al., 2014b*). To establish the specific temporal requirements of α'2 MBON and determine whether its corresponding α2α'2 dopaminergic input is necessary for alcohol-associated preference, we thermogenetically inactivated neurotransmission during either acquisition or retrieval. Inactivating α'2 cholinergic MBONs or its corresponding α2α'2 DANs during retrieval, but not acquisition, significantly reduced alcohol-associated preference (*Figure 5C–F*). The involvement of α2α'2 DANs is particularly interesting because it demonstrates a requirement of a separate population of DANs in memory expression.

Interestingly, *trans*-Tango did not identify the α'2 cholinergic MBON as a postsynaptic target of α2α'2 DANs. Of course, the possibility exists that there remains connectivity not identified by *trans*-Tango, however, RNAi against D1Rs or D2Rs did not disrupt alcohol-associated preference (*Figure 5—figure supplement 1*), suggesting that, like the β'2 microcircuit necessary for retrieval of alcohol-associated memories, direct connectivity of the α'2 microcircuit is not required for alcohol-associated preference.

## Alcohol memory expression circuits converge on a higher-order integration center

Emerging models in the MB field suggest that MBON activity is pooled across compartments and that learning shifts the balance of activity to favor approach or avoidance (*Owald and Waddell, 2015*). It remains unclear where this MBON activity converges. In order to identify potential regions that integrated MBON activity, we used *trans*-Tango to map postsynaptic partners of α'2, β'2mp, and β2β'2a MBONs. Interestingly, the dorsal regions of the FSB, specifically layers 4/5 or layer 6, were consistently identified as postsynaptic targets of α'2 MBON (*Figure 6a,c*). Both β'2mp and β2β'2a MBONs also have synaptic connectivity with the dorsal regions of the FSB (*Figure 6b,d*). Together these data reveal the dorsal FSB as an intriguing convergent region downstream of the MB whose role in alcohol-associated preference should be investigated further (*Figure 6e*).

## Discussion

In this study we provide novel insight to the circuit-level mechanisms underlying the acquisition and expression of alcohol reward memories in *Drosophila*. We found that acquisition of appetitive response for alcohol does not rely on subsets of DANs, but instead requires population level dopaminergic modulation of the MB via PAM DANs, which increases with prolonged exposure (*Figure 7A*). The expression of alcohol reward memories, however, requires two discrete dopamine microcircuits within the vertical and horizontal lobes, which converge at several points: a neuron that regulates memory consolidation and the dorsal layers of the FSB (*Figure 7B*). We hypothesize that

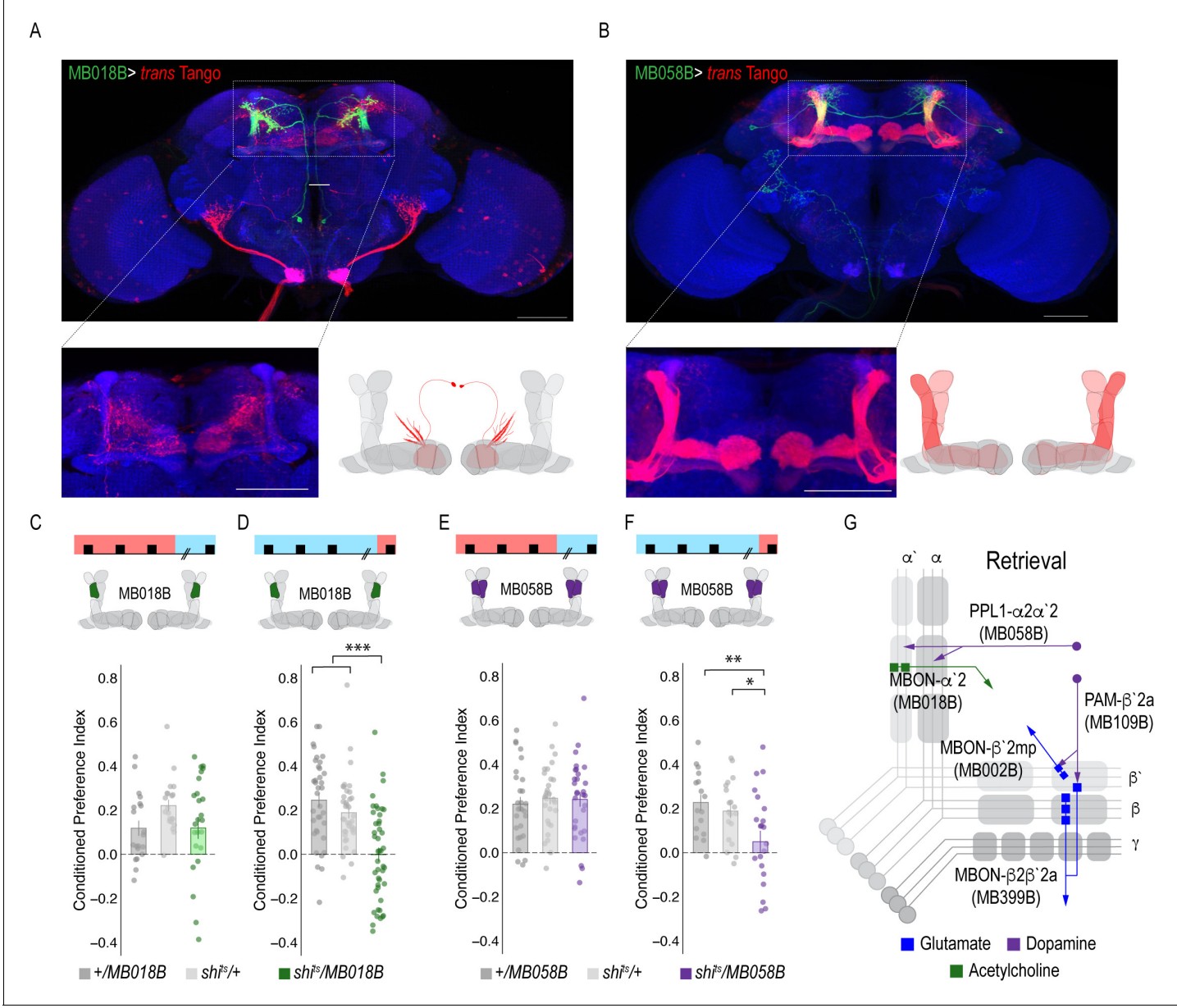

**Figure 5.** A microcircuit within the vertical lobe is important for alcohol-associated preference. (**A**) Representative maximum projection confocal stacks of MB018B > *trans*-Tangorevealed that β'2mp glutamatergic MBON is postsynaptic to α'2 cholinergic neuron (**B**) Representative maximum projection confocal stacks of MB058B > *trans*-Tangosuggests that that α'2 cholinergic MBON is not postsynaptic α2α'2 DAN (**C**) Thermogenetic inactivation of α'2 cholinergic neurons during acquisition did not affect the expression of alcohol-associated preference F(2,63)=2.18, p=0.12. (**D**) Inactivation of α'2 cholinergic neurons during retrieval significantly reduced preference F(2,116)=19.46, p=5.17×10–08. (**E**) Similarly, thermogenetic inactivation of α2α'2 DANs during acquisition did not affect the expression of alcohol-associated preference F(2,85)=0.202, p=0.817. (**F**) Inactivation during retrieval significantly reduced preference F(2,54)=5.103, p=0.009. (**G**) Updated model of circuits responsible for the expression of alcohol-associated preference during retrieval. Scale bar = 50 μm.

The online version of this article includes the following figure supplement(s) for figure 5:

**Figure supplement 1.** Decreasing expression of dopamine receptors in the α'2 MBON did not affect alcohol-associated preference.

**Figure supplement 2.** Temperature controls for lines that showed decreases in retrieval of alcohol-associated preference at a restrictive temperature.

these convergent points provide multiple opportunities for memory to be updated or strengthened to influence subsequent behavior.

Surprisingly, contrary to adaptive aversive or appetitive memories in flies (*Liu et al., 2012*; *Yamagata et al., 2016*; *Yamagata et al., 2015*; *Masek et al., 2015*), encoding alcohol-associated

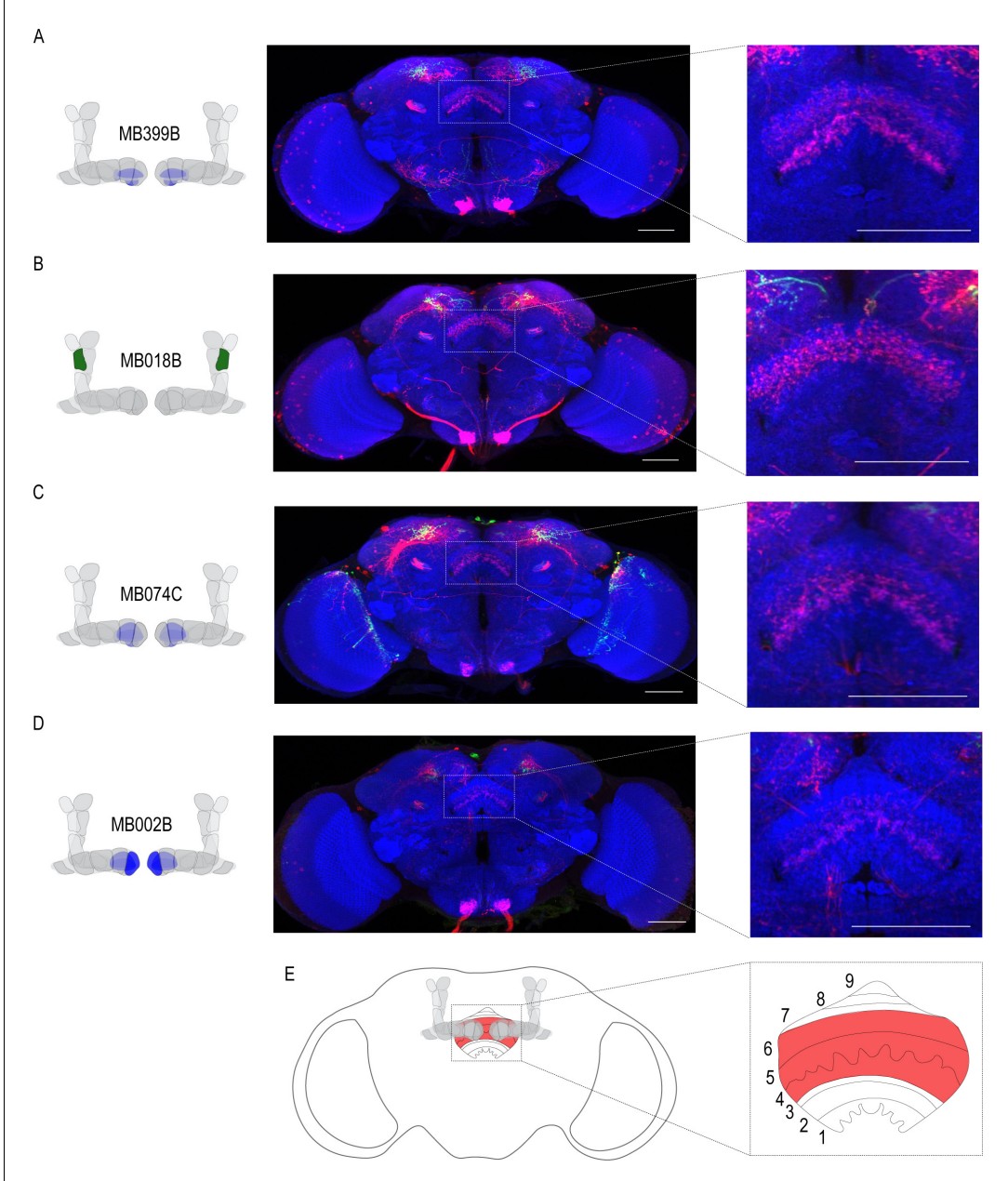

**Figure 6.** Circuits important for memory expression at retrieval converge on the dorsal FSB. (**A**) Confocal stack of FSB highlighting the postsynaptic signal of β2β'2a MBON in the FSB. This MBON predominately targets layers 4, and 6. (**B**) Confocal stack of FSB highlighting the postsynaptic signal of α'2 MBON in the FSB. This MBON predominately targets layer 6. (**C–D**) Confocal stack of FSB highlighting the postsynaptic signal of β'2mp MBON in the FSB. This MBON predominately targets layer 4 and 5. (**E**) Schematic of the fly brain highlighting the FSB and its layers. The FSB is a 9-layer structure (**Wolff et al., 2015**), of which 4,5, and 6 are targets. Scale bar = 50 μm.

preference is not dependent on a single subset of DANs or MBON. Instead, acquisition appears to depend on a population of DANs whose activity emerges over the course of exposure to intoxicating doses of alcohol and likely increase across odor-alcohol pairing sessions via the recruitment of neurons. Although we cannot rule out the influence of other neurotransmitters or peptides that are potentially co-released with dopamine, dopamine auto receptor knock-down experiments in PAM neurons using the *R58E02-GAL4* driver suggests that the regulation of dopamine release at the synapse is important for alcohol reward memory.

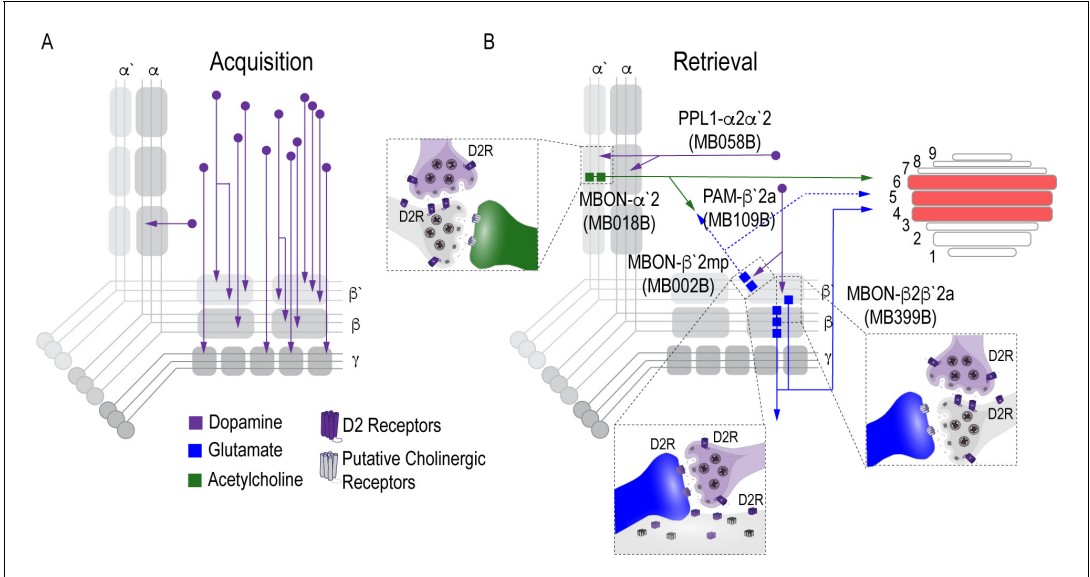

**Figure 7.** Proposed circuits important for the acquisition and expression of alcohol-associated preference. (**A**) The proposed acquisition circuit comprises population activity from DANs innervating the MB (**B**) The proposed retrieval circuit comprises two separate microcircuits, one in the vertical lobe and one in the horizontal lobe that converge on β'2mp MBON as well as layers with the dorsal FSB. The vertical lobe microcircuit includes a PPL1 α'2α2 DAN that have indirect connections with an α'2 cholinergic MBON via the MB. The horizontal lobe microcircuit includes a subset of PAM DANs (β'2a) that have direct connections with the β'2mp glutamatergic MBON and indirect connections with the β2β'2a glutamatergic MBON. Vertical and horizontal microcircuits converge on the β'2mp glutamatergic MBON which is important for arousal (*Sitaraman et al., 2015*) and layers 4, 5, and 6 of the FSB.

Previous work in *Drosophila* reports that increasing the number of encoding DANs enhances how long aversive memory lasts (*Aso and Rubin, 2016*). Remarkably in an independent set of similar experiments, *Ojelade et al., 2019* demonstrate that previous alcohol exposure potentiates dopaminergic responses to subsequent artificial activation. Together these findings are consistent with what is reported in mammalian models, where most drugs of abuse initially increase dopamine levels beyond what is experienced during natural reward (*Nutt et al., 2015*; *Volkow and Morales, 2015*; *Kegeles et al., 2018*) and suggest a general rule where stability of memory is encoded by the number of DANs involved during acquisition. We hypothesize that the recruitment of additional DANs and the potentiation of their responses across sessions contributes to the stability of alcohol memory. Understanding the mechanism by which DANs are recruited may provide powerful insight into why memories for an intoxicating experience are so persistent.

Surprisingly, despite the involvement of α1 PAM DANs in the acquisition of long-term sucrose reward memory (*Ichinose et al., 2015*), the α1 DANs do not appear to play a role in alcohol- associated preference. Perhaps differences in the animal's internal state and/or temporal dynamics of alcohol intoxication underlies the distinction in requisite circuits. It's possible that the involvement of α1 is limited to internal states of hunger and thus not required when flies are sated. Unlike long-term sucrose memory, alcohol-reward memory is present in both hungry and sated flies, offering a unique opportunity to study how internal state might influence circuit selection for memory expression. Further investigation and comparison of circuits important for alcohol-reward memory in hungry, sated, and other internal states should prove to be a compelling line of research.

Systems memory consolidation suggests that there are different circuits for memory acquisition and expression. Indeed, work in both fly and mammalian models suggest brain regions have a time-limited role in systems consolidation (*Trannoy et al., 2011*; *Zars et al., 2000*; *Blum et al., 2009*; *Akalal et al., 2011*; *Qin et al., 2012*; *Cervantes-Sandoval et al., 2013*; *Krashes et al., 2007*; *Krashes and Waddell, 2008*; *Perisse et al., 2013*; *Roy et al., 2017*). Our data suggest that population encoding during acquisition shifts to sparse representation during memory expression and distinct processes regulate consolidation and expression. The expression of alcohol-associated preference is dependent on two separate microcircuits defined by a small subset of PAM DANs

(β'2a) within a larger population of reward encoding DANs and a single paired posterior lateral (PPL1; α2α'2) DAN (*Figure 7B*). Additionally, we found β'2a DANs make direct connections with a glutamatergic MBON (β'2mp) implicated in arousal (*Sitaraman et al., 2015*). Converging microcircuits emerge with time, and are not necessary for the acquisition of these long-lasting preference associations (*Figure 7B*). Interestingly, blocking β'2mp MBON when flies normally sleep enhanced memory in a D2R-dependent manner. We propose that β'2a DANs inhibit β'2mp MBONS neuronal activity, thus permitting consolidation of alcohol-associated preference.

The involvement of PAM β'2a DANs in the expression of alcohol-associated preference is particularly interesting because these neurons (targeted by broader driver lines *104* Gal4 and *R48B04-Gal4*) were previously implicated in the acquisition of 3 min sucrose memory in starved animals (*Burke et al., 2012*), as well as naïve water seeking in thirsty animals (*Lin et al., 2014*). β'2a DANs were also previously reported to inhibit β'2amp MBONs to promote approach behaviors when flies were presented with conflicting aversive and appetitive odor cues (*Lewis et al., 2015*). The effects of β'2a dopamine neuronal inhibition, however, were not long lasting. Instead, the appetitive food odor, and consequently the activity of β'2a DANs, appears to act as an occasion setter, or a discriminatory stimulus that augments an animal's response to a cue (*Lewis et al., 2015*). We speculate this neuron resets the response to a cue associated with alcohol, which may be critical for overcoming the initial aversive properties of alcohol. The involvement of PPL1 a2a'2 DANs are also interesting because PPL1 DANs are typically responsible for assigning negative valences to associated cues (*Aso et al., 2012*; *Waddell, 2013*; *Claridge-Chang et al., 2009*; *Kim et al., 2018*; *Boto et al., 2019*), suggesting that a microcircuit associated with negative valence directly interacts with a microcircuit associated with positive valence to regulate the decision to seek alcohol. We hypothesize that repeated intoxicating experiences change the dynamics of β'2a DANs during acquisition or consolidation in a way that creates long term changes to the responsivity of the β'2mp MBON, perhaps to the α'2 MBON.

Because the β'2mp MBON is not required for expression of memory, it is likely that its output is integrated elsewhere in the brain to drive goal directed behaviors. Indeed, there is a wealth of examples in the literature of the systems balancing input from integrating neural circuits to drive goal directed behavior (*Buschman and Miller, 2014*; *Hoke et al., 2017*; *Knudsen, 2007*; *Perisse et al., 2013*; *Owald et al., 2015*; *Owald and Waddell, 2015*; *Aso et al., 2014b*; *Lewis et al., 2015*; *Dolan et al., 2018*). Here we have identified one such structure: the dorsal layers of the FSB, specifically layers 4, 5, and 6, that is an anatomical candidate for pooling MB output activity to drive learned behaviors. Interestingly, although the FSB has an established role in arousal and sleep, more recent work has defined its role in innate and learned nociceptive avoidance further supporting its role in integrating MBON activity (*Hu et al., 2018*). We hypothesize that signals from the β2β'2a and α'2 MBONs are integrated at the FSB to shift naïve response to cue-directed learned response. Compellingly, the β'2mp MBON, which we show is required for consolidation of alcohol-associated preference, also sends projections the FSB. This presents a circuit framework through which memory could be updated to influence behavioral expression. There are likely other convergent and or downstream structures that are important for reward processing and the emerging full connectome will better shed light on these connections.

Alcohol is a unique stimulus, because unlike natural rewards and punishments, it has both aversive and appetitive properties. Flies naively will avoid intoxicating doses of alcohol, but avoidance switches to preference with experience (*Shohat-Ophir et al., 2012*; *Peru Y Colón de Portugal et al., 2014*; *Ojelade et al., 2019*; *Kaun et al., 2011*). Previous work in starved flies have similarly described the formation of parallel competing memories when rewards are tainted with bitter tastants (*Das et al., 2014*). In this case, cue-associated avoidance switches to approach around the same time that the nutritional value of sugar is processed (*Musso et al., 2015*; *Das et al., 2014*). During memory acquisition, both bitter taste and shock memories require the MP1 DA neuron, whereas sucrose memories, like alcohol memories, require the PAM neurons. Similar to our work, *Ojelade et al., 2019* show that the PAM population of DANs projecting to the MB are required for acquisition of experience-dependent alcohol preference in a consumption assay. They also demonstrate that activating layer six of dorsal FSB leads to naïve alcohol preference. These data are particularly exciting because we also identified the dorsal FSB as a convergent structure to MBONs important for the consolidation and expression of alcohol-associated preference. Perhaps the

temporal nature of a valence switch from conditioned aversion to preference is a consequence of system level interactions between the MB and FSB.

A classic hallmark of addiction is the enduring propensity to relapse, which is often driven by drugs associated cues. We believe our work provides valuable insight to the mechanisms by which drugs of abuse regulate acquisition, consolidation, and expression of pervasive sensory memories. Here we establish a circuit framework for studying the neural mechanisms of alcohol reward memory persistence in *Drosophila* and understanding how circuits change in drug-induced states.

# Materials and methods

## Key resources table

| Reagent type (species) or resource | Designation | Source or reference | Identifiers | Additional information |
|---|---|---|---|---|
| Genetic reagent (*D. melanogaster*) | y[1]w[*] | *Pfeiffer et al., 2008* | | |
| Genetic reagent (*D. melanogaster*) | UAS-shibire*ts1* | *Pfeiffer et al., 2012* | FLYB: FBst0066600; RRID:BDSC_66600 | |
| Genetic reagent (*D. melanogaster*) | UAS-GCaMP6m | | FLYB: FBst0042750; RRID:BDSC_42750 | FlyBase symbol: Dmel\PBac{20XUAS-IVS-GCaMP6m}VK00005 |
| Genetic reagent (*D. melanogaster*) | UAS-Dop2R-RNAi | *Dietzl et al., 2007* | FLYB: FBsf0000079969; VDRC ID: 11471 | FlyBase symbol: Dmel\dsRNA-GD11471 (currently unavailable) |
| Genetic reagent (*D. melanogaster*) | UAS-Dop1R1-RNAi | *Keleman et al., 2009*) | FLYB: FBsf0000090794; dsRNA-KK107258 VDRC ID: 100249 | FlyBase symbol: Dmel\dsRNA-KK107258 |
| Genetic reagent (*D. melanogaster*) | UAS-Dop1R2-RNAi | *Dietzl et al., 2007* | FLYB: FBsf0000073893 dsRNA-GD3391 VDRC ID: 3391 | FlyBase symbol: Dmel\dsRNA-GD3391 (currently unavailable) |
| Genetic reagent (*D. melanogaster*) | trans-Tango: UAS-myrGFP, QUAS-mtdTomato (3xHA); brp-SNAP | *Kohl et al., 2014*, *Talay et al., 2017* | | |
| Genetic reagent (*D. melanogaster*) | R58E02-Gal4 | *Liu et al., 2012* | FLYB: FBtp0061564; RRID:BDSC_41347 | Flybase Symbol: P{GMR58E02-GAL4} |
| Genetic reagent (*D. melanogaster*) | R58E02; mcD8::GFP | this paper | FLYB: FBtp0061564; RRID:BDSC_41347 FLYB: FBti0200979; RRID:BDSC_79626, | Flybase Symbol: P{GMR58E02-GAL4} Flybase Symbol: Dmel\P{10XUAS-mCD8GFP-APEX2}attP40 |
| Genetic reagent (*D. melanogaster*) | HL9-Gal4 | *Claridge-Chang et al., 2009* | FLYB: FBtp0073020 RRID:BDSC_69208 | Flybase Symbol: P{HL9-GAL4.DBD}attP2 |
| Genetic reagent (*D. melanogaster*) | MB058B-split-Gal4 | *Aso et al., 2014a* | FlyLight Robot ID: 2135107 | |
| Genetic reagent (*D. melanogaster*) | MB109B-split-Gal4 | *Aso et al., 2014a* | FlyLight Robot ID: 2135157 | |
| Genetic reagent (*D. melanogaster*) | MB040B-split-Gal4 | *Aso et al., 2014a* | N/A | |
| Genetic reagent (*D. melanogaster*) | MB042B-split-Gal4 | *Aso et al., 2014a* | FlyLight Robot ID: 2135092 | |
| Genetic reagent (*D. melanogaster*) | MB188B-split-Gal4 | *Aso et al., 2014a* | FlyLight Robot ID: 2135236 | |
| Genetic reagent (*D. melanogaster*) | MB032B-split-Gal4 | *Aso et al., 2014a* | FlyLight Robot ID: 2135083 | |
| Genetic reagent (*D. melanogaster*) | MB087C-split-Gal4 | *Aso et al., 2014a* | FlyLight Robot ID: 2135135 | |
| Genetic reagent (*D. melanogaster*) | MB301B-split-Gal4 | *Aso et al., 2014a* | FlyLight Robot ID: 2135349 | |

*Continued on next page*

*Continued*

| Reagent type (species) or resource | Designation | Source or reference | Identifiers | Additional information |
|---|---|---|---|---|
| Genetic reagent (*D. melanogaster*) | *MB002B-split-Gal4* | *Aso et al., 2014a* | FlyLight Robot ID:2135053 | |
| Genetic reagent (*D. melanogaster*) | *MB011B- split-Gal4* | *Aso et al., 2014a* | FlyLight Robot ID:2135062 | |
| Genetic reagent (*D. melanogaster*) | *MB-210B-split-Gal4* | *Aso et al., 2014a* | FlyLight Robot ID: 2135258 | |
| Genetic reagent (*D. melanogaster*) | *MB018B-split-Gal4* | *Aso et al., 2014a* | FlyLight Robot ID: 2135069 | |
| Genetic reagent (*D. melanogaster*) | *MB399B-split-Gal4* | *Aso et al., 2014a* | FlyLight Robot ID: 2501738 | |
| Genetic reagent (*D. melanogaster*) | *MB074C-split-Gal4* | *Aso et al., 2014a* | FlyLight Robot ID: 2135112 | |
| Antibody | α-GFP (Rabbit polyclonal) | Life Tech | Cat#A11122 | (1:1000) |
| Antibody | α-HA (Rat monoclonal) | Roche | Cat#11867423001 | (1:100) |
| Antibody | Goat α-Rabbit AF488 | Life Tech | Cat#A11034 | (1:400) |
| Antibody | Goat α-Rat AF568 | Life Tech | Cat#A11077 | (1:400) |
| Antibody | α-dopamine (Mouse monoclonal) | Millipore | Cat#MAB5300 | (1:40) |
| Antibody | Goat α-mouse AF488 | Thermo | Cat#A11029 | (1:200) |
| Sequence-based reagent | CG13646F | *Petruccelli et al., 2018* | PCR primers | AGTTTGACATCCACCCCGTC |
| Sequence-based reagent | CG13646R | *Petruccelli et al., 2018* | PCR primers | CTCACTGGCGATTCCGATGA |
| Sequence-based reagent | Dop2RF | *Petruccelli et al., 2018* | PCR primers | CTGAACTGCACCAACGAGACGC |
| Sequence-based reagent | Dop2RR | *Petruccelli et al., 2018* | PCR primers | CAGGATGTTGCCGAAGAGGGTC |
| Sequence-based reagent | Dop1R1F | this paper | PCR primers | CCGTCGTGTCCAGCTGTATCAG |
| Sequence-based reagent | Dop1R1R | this paper | PCR primers | CTTCTCGGCCACCTCACCTG |
| Sequence-based reagent | Dop1R2F | this paper | PCR primers | CCTGGCTCGGCTGGATCAAC |
| Sequence-based reagent | Dop1R2R | this paper | PCR primers | ATCGTGGGCTGGTACTTGCG |

## Fly strains

All *Drosophila melanogaster* lines were raised on standard cornmeal-agar media with tegosept anti-fungal agent and maintained at either 18C or 21C. For a list of fly lines used in the study, see Key Resources Table. All *Drosophila melanogaster* lines used for *trans-Tango* were raised and maintained at 18C in humidity-controlled chambers under 14/10 hr light/dark cycles on standard cornmeal-agar media with tegosept anti-fungal agent.

## Behavioral experiments

### Odor preference conditioning

For behavior experiments, male flies were collected 1–2 days post eclosion and were shifted from 21C to 18C, 65% humidity and placed on a 14/10 hr light/dark cycle. Odor conditioning was performed similar to *Kaun et al., 2011*. In short, groups of 30 males were trained in perforated 14 ml culture vials filled with 1 ml of 1% agar and covered with mesh lids. Training rooms were temperature and humidity controlled (65%). Training was performed in the dark with minimal red-light

illumination and was preceded by a 20 min habituation to the training chambers. Training chambers were constructed out of PlexiGlas (30 × 15×15 cm) (for details please refer to *Nunez et al., 2018*). During habitation, humidified air (flow rate: 130) was streamed into the chambers. A single training session consisted of a 10 min presentation of odor 1 (flow rate: 130), followed by a 10 min presentation of odor 2 (flow rate 130) with 60% ethanol (flow rate 90: ethanol 60: air). Reciprocal training was performed simultaneously to ensure that inherent preference for either odor did not affect conditioning scores. For the majority of experiments odors used were 1:36 isoamyl alcohol and 1:36 isoamyl acetate in mineral oil, however, screen behavioral experiments used 1:36 isoamyl alcohol and 1:36 ethyl acetate in mineral oil. Vials of flies from group one and group two were age matched and paired according to placement in the training chamber. Pairs were tested simultaneously 24 hr later in the Y maze by streaming odor 1 and odor 2 (flow rate 10) in separate arms and allowing flies to walk up vials to choose between the two arms. A preference index was calculated by # flies in the paired odor vial- # flies in the unpaired odor vial)/total # of flies that climbed. A conditioned preference index (CPI) was calculated by the averaging preference indexes from reciprocal groups. All data are reported as CPI. All plots were generated in RStudio.

## Odor sensitivity

Odor sensitivity was evaluated at restrictive temperatures (30°C). Odors used were 1:36 isoamyl alcohol in mineral oil and 1:36 isoamyl acetate in mineral oil. Groups of 30 naïve males were presented with either an odor (flow rate 10) or air streamed through mineral oil in opposite arms of the Y. Preference index was calculated by # flies in odor vial- # flies in air vial)/total # flies that climbed for each individual odor.

## Ethanol sensitivity

Ethanol sensitivity was evaluated in the recently developed flyGrAM assay (*Scaplen et al., 2019*). Briefly, for thermogenetic inactivation, 10 flies were placed into arena chambers and placed in a 30° C incubator for 20 min prior to testing. The arena was then transferred to a preheated (30°C) light sealed box and connected to a vaporized ethanol/humidified air delivery system. Flies were given an additional 15 min to acclimate to the box before recordings began. Group activity was recorded (33 frames/sec) for five minutes of baseline, followed by 10 min of ethanol administration and five minutes of following ethanol exposure. Activity was binned by 10 s and averaged within each genotype. Mean group activity is plotted as a line across time with standard error of the mean overlaid. All activity plots were generated in RStudio. *trans*-Tango immunohistochemistry and microscopy.

Experiments were performed according to the published FlyLight protocols with minor modifications. Briefly, either adult flies that are 15–20 days old were cold anaesthetized on ice, de-waxed in 70% ethanol dissected in cold Schneider's Insect Medium (S2). Within 20 min of dissection, tissue was incubated in 2% paraformaldehyde (PFA) in S2 at room temperature for 55 min. After fixation, samples were rinsed with phosphate buffered saline with 0.5% Triton X-100 (PBT) and washed 4 times for 15 min at room temperature. Following PBT washes, PBT was removed and samples were incubated in SNAP substrate diluted in PBT (SNAP-Surface649, NEB S9159S; 1:1000) for 1 hr at room temperature. Samples were then rinsed and washed 3 times for 10 min at room temperature and then blocked in 5% heat-inactivated goat serum in PBT for 90 min at room temperature and incubated with primary antibodies (Rabbit α-GFP Polyclonal (1:1000), Life Tech #A11122, Rat α-HA Monoclonal (1:100), Roche #11867423001) for two overnights at 4°C. Subsequently, samples were rinsed and washed 4 times for 15 min in 0.5% PBT and incubated in secondary antibodies (Goat α-Rabbit AF488 (1:400), Life Tech #A11034, Goat α-Rat AF568 (1:400), Life Tech #A11077) diluted in 5% goat serum in PBT for 2–3 overnights at 4°C. Samples were then rinsed and washed 4 times for 15 min in 0.5% PBT at room temperature and prepared for DPX mounting. Briefly, samples were fixed a second time in 4% PFA in PBS for 4 hr at room temperature and then washed four times in PBT at room temperature. Samples were rinsed for 10 min in PBS, placed on PLL-dipped cover glass, and dehydrated in successive baths of ethanol for 10 min each. Samples were then soaked three times in xylene for 5 min each and mounted using DPX. Confocal images were obtained using a Zeiss, LSM800 with ZEN software (Zeiss, version 2.1) with auto Z brightness correction to generate a homogeneous signal where it seemed necessary, and were formatted using Fiji software (http://fiji.sc).

## Dopamine immunohistochemistry and microscopy

Groups of flies were exposed to either 10 min of air or 10 min of ethanol and dissected within 15 min of exposure on ice. Immunohistochemistry was performed according to *Cichewicz et al., 2017*. With 15 min of dissection, tissue was transferred to fix (1.25% glutaraldehyde in 1% PM) for 3–4 hr at 4°C. Tissue was subsequently washed 3 times for 20 min in PM and reduced in 1% sodium borohydride. Then the tissue was washed 2 times for 20 min before a final wash in PMBT. Tissue was blocked in 1% goat serum in PMBT overnight at 4°C and incubated in primary antibody (Mouse anti-dopamine (1:40) Millipore Inc, #MAB5300) for 48 hr at 4°C. Following primary antibody incubation, tissue was washed three times in PBT for 20 min at room temperature and incubated in secondary antibody (Goat anti mouse 488 (1:200 in PBT) Thermo #A11029) for 24 hr at 4°C. The following day tissue was washed 2 times for 20 min in PBT and then overnight in fresh PBT. Tissue was rinsed quickly in PBS, cleared in FocusClear and mounted in MountClear (Cell Explorer Labs). Confocal images were obtained using a Zeiss, LSM800 with ZEN software (Zeiss, version 2.1). Microscope settings were established using ethanol tissue before imaging air and ethanol samples.

## Dopamine fluorescence analysis

Fluorescence was quantified in Fiji (*Schindelin et al., 2012*) using Segmentation Editor and 3D Manager (*Ollion et al., 2013*). In segmentation editor ROIs were defined using the selection tool brush to outline the MB in each slice and also outside a background region immediately ventral to MB that lacked defined fluorescent processes. 3D ROIs of the MB and control region were created by interpolating across slices. Geometric and intensity measurements were calculated for each ROI in 3D Manager and exported to CSV files. Integrated density for each ROI was normalized by the integrated density of control regions. Average integrated density for air and ethanol exposures are reported. All fluorescence quantifications were performed by a blinded experimenter.

## Calcium imaging protocol and analysis

To express GCaMP6m in PAM neurons, UAS-GCaMP6m virgin female flies were crossed to male flies containing the R58E02-GAL4 driver. As previously mentioned, all flies were raised on standard cornmeal-agar food media with tegosept anti-fungal agent and maintained on a 14/10 hr light/dark cycle at 24°C and 65% humidity.

### Fly Preparation

Male flies were selected for imaging six days post-eclosion. Flies were briefly anesthetized on ice to transfer and fix to an experimental holder made out of heavy-duty aluminum foil. The fly was placed into an H-shaped hold cut out of the foil and glued in place using epoxy (5 min Epoxy, Devcon). The head was tilted about 70° to remove the cuticle from the back of the fly head. All legs were free to move, the proboscis and antenna remained intact and unglued. Once the epoxy was dry, the holder was filled with *Drosophila* Adult Hemolymph-Like Saline (AHLS). The cuticle was removed using a tungsten wire (Roboz Surgical Instruments Tungsten Dissecting Needle,. 125 mm, Ultra Fine) and forceps #5. The prepared fly in its holder was positioned on a customized stand underneath the two-photon scope. The position of the ball and the stream delivery tubes were manually adjusted to the fly's position in the holder.

### Imaging paradigm

Calcium imaging recordings were performed with a two-photon resonance microscope (Scientifica). Fluorescence was recorded from the PAM neurons innervating the mushroom body for a total duration of 80 to 95 min. The first 10 min the fly was presented an air stream, followed by 10 min of isoamyl alcohol. The fly was then presented with 10 min of isoamyl alcohol paired with ethanol followed by 50 min of streaming air. To avoid bleaching effects and to match the higher resolution imaging properties, the recording was not throughout the entire paradigm but spaced with imaging intervals of 61.4 s. Recordings were performed using SciScan Resonance Software (Scientifica). The laser was operated at 930 nm wavelength at an intensity of 7.5–8 mW. Images were acquired at 512 × 512 pixel resolution with an average of 30.9 frames per second. Recordings lasted 1900 frames which equals 61.5 s. Recordings were performed at 18.5°C room temperature and 59% humidity.

## Imaging analysis

Data were registered, processed, and extracted using a matlab GUI developed by C. Deister, Brown University. Calcium image files (.tiff) comprising of 1900 frames taken at 30.94 frames per second rate (61.4 s), were initially averaged every five frames to downsize the. tiff image files to 380 frames. Image files were then aligned and registered in X-Y using a 15–50 frame average as a template. ROIs were constructed over the MB lobes using non-negative matrix factorization to identify active regions and then subsequently segmented to create the ROIs. Fluorescence values were extracted from identified ROIs and $\Delta F/F_o$ measurements were created using a moving-average of 75 frames to calculate the baseline fluorescence ($F_o$). Average fluorescence traces across flies (n = 6) were visualized using ggplot in R Studio. Fiji (*Schindelin et al., 2012*) was used to construct heat-maps visualizing calcium activity. Calcium image files were summated across 1900 frames to create Z-projections. A heat gradient was used to visualize calcium activity magnitude. qRT-PCR qRT-PCR methods have been described previously (*Petruccelli et al., 2018*). In brief, total RNA was extracted from approximately 100 heads using Trizol (Ambion, Life Technologies) and treated with DNase (Ambion DNA-Free Kit). Equal amounts of RNA (1 µg) were reverse-transcribed into cDNA (Applied Biosystems) for each of the samples. Then, Biological (R3) and technical (R2) replicates were analyzed with Sybr Green Real-Time PCR (BioRad, ABI PRISM 7700 Sequence Detection System) performed using the following PCR conditions: 15 s 95℃, 1 min 55℃, 40x. Primer sequences can be found in *Supplementary file 1— Table 4*. Across all samples and targets, Ct threshold and amplification start/stop was set to 0.6 and manually adjusted, respectively. All target genes were initially normalized to *CG13646* expression for comparative $\Delta$Ct method analysis, then compared to control genotype to assess fold enrichment ($\Delta\Delta$ Ct method).

## Acknowledgements

This work was supported by the Smith Family Award Program for Excellence in Biomedical Research (KRK), the Israel Binational Science Foundation Start Up Grant 2015005 (KRK in collaboration with Moshe Parnas, University of Tel Aviv), the Rhode Island Foundation Medical Research Fund 20144133 (KMS and KRK), NIH grant R01AA024434 (KRK) and Brown University OVPR Seed Funding (KRK and GB). KRK was also supported by a grant from NIH (5P20GM103645-03) to the Carney Institute for Brain Science Center for Nervous System Function COBRE. GB and MT were supported in part by NIH grants 1R01DC17146 and 5R01MH105368. We thank Adrian Rothenfluh (University of Utah) for ongoing dialogue of this work, and all of the Kaun lab members for fruitful discussions and feedback on earlier versions of this manuscript. We also thank the *Drosophila* community, particularly Yoshinori Aso (Janelia Research Campus) and Gerry Rubin (Janelia Research Campus), the Bloomington Stock Center, and the Vienna *Drosophila* RNAi Center for sharing fly stocks. Y Aso also provided confocal stacks of MB042B and MB040B used for cell counting. We thank Thomas Boudier (Université Pierre et Marie Curie, Sorbonne Universités) and Stefan Helfrich (Bioimaging Center of the University of Konstanz) for helpful software advice, Jay Hirsh (University of Virginia) for reagents and advice, and Christopher Diester (Brown University) and other members of Chris Moore's lab (Brown University) for calcium analysis guidance. Finally, we thank John McGeary (Brown University), Tara White (Brown University), and Daniel Dombeck (Northwestern University) for helpful comments on an earlier version of this manuscript.

## Additional information

### Funding

| Funder | Grant reference number | Author |
| --- | --- | --- |
| National Institute on Alcohol Abuse and Alcoholism | R01AA024434 | Karla R Kaun |
| National Institute of General Medical Sciences | P20GM103645 | Karla R Kaun |
| Smith Family Award Program for Excellence in Biomedical Research | | Karla R Kaun |

| | | |
|---|---|---|
| Binational Science Foundation | 2015005 | Karla R Kaun |
| Rhode Island Foundation Medical Research Fund | 20144133 | Kristin M Scaplen Karla R Kaun |
| Brown University Seed Funding | | Gilad Barnea Karla R Kaun |
| National Institute on Deafness and Other Communication Disorders | R01DC17146 | Gilad Barnea Mustafa Talay |
| National Institute of Mental Health | R01MH105368 | Gilad Barnea Mustafa Talay |

The funders had no role in study design, data collection and interpretation, or the decision to submit the work for publication.

### Author contributions

Kristin M Scaplen, Conceptualization, Data curation, Formal analysis, Validation, Investigation, Visualization, Methodology; Mustafa Talay, Resources, Data curation; Kavin M Nunez, Data curation, Formal analysis; Sarah Salamon, Amanda G Waterman, Sydney Gang, Sophia L Song, Data curation; Gilad Barnea, Resources; Karla R Kaun, Conceptualization, Resources, Supervision, Funding acquisition, Investigation, Visualization, Methodology, Project administration

### Author ORCIDs

Kristin M Scaplen (iD) https://orcid.org/0000-0001-7493-1420
Gilad Barnea (iD) https://orcid.org/0000-0001-6842-3454
Karla R Kaun (iD) https://orcid.org/0000-0002-8756-9528

### Decision letter and Author response

Decision letter https://doi.org/10.7554/eLife.48730.sa1
Author response https://doi.org/10.7554/eLife.48730.sa2

## Additional files

### Supplementary files

• Supplementary file 1. Supplemental Tables that accompany data from *Figures 1*, *2*, *3*, *4* and *5*. Table 1 includes a cell count of PAM neuron driver lines. Table 2 includes odor sensitivity controls for all memory experiments performed. Table 3 includes a description of target and off-target expression of split-Gal4 lines used. Table 4 includes a comprehensive table of detailed statistics that describe all data. Table 5 includes a review of papers published that include use of the RNAi lines used here.

• Transparent reporting form

### Data availability

All data generated or analysed during this study are included in the manuscript and supporting files.

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
