## [Decision Letter]

**Acceptance summary:**

This manuscript expands on prior circuit ideas for ethanol reward learning and memory in *Drosophila*. Prior to this manuscript, we knew that distinct groups of dopamine neurons (PAM and non-PAM), the mushroom body intrinsic neurons, and a variety of mushroom body output neurons (MBONs) were required for alcohol memory. This manuscript delineates the individual neurons required, and assigns their function to acquisition, consolidation, and retrieval, providing circuit-level understanding. This paradigm, pioneered in the Kaun lab, is one of the best ways to mechanistically address the effects of alcohol intake on behavior, and this paper advances our understanding of these processes.

**Decision letter after peer review:**

Thank you for submitting your article "Circuits that encode and predict alcohol-associated preference" for consideration by *eLife*. Your article has been reviewed by three peer reviewers, and the evaluation has been overseen by K VijayRaghavan as the Senior Editor and Leslie Griffith as the Reviewing Editor. The reviewers have opted to remain anonymous.

The reviewers have discussed the reviews with one another and the Reviewing Editor has drafted this decision to help you prepare a revised submission.

Essential revisions:

1) There are many problems with the way the paper is written. The references to the mammalian literature on dopamine function in Results are unnecessary and potentially misleading – there were no experiments done with mice or rats. The authors make comparisons or claims about many different areas of the addiction literature. If the authors wish to defend this breadth, then more accurate assessment of their experiments in relation to the areas of the literature is needed. For example, the prediction error literature in mammals is grounded in electrophysiological measurements and assessments of populations of neurons, whereas the experiments presented here are inactivation by dominant negative Shibire in genetically tagged neurons. Further, there is no information about whether the fly neurons are permissive or instructive for the behaviors, we do not know if they are firing action potentials, and if they are, we do not know if the firing is time locked to the cue, ethanol, or some aspect of the behavioral sequence. At minimum, the discussion of the scientific background and link to mammalian literature should all be moved to the Discussion. In addition, the text is unnecessarily long and can be more concise, especially for Figure 2, where single neurons have not actually been identified. Authors may also want to minimize repetitions, as this lengthens the paper unnecessarily (e.g. subsections “A dopamine-glutamate circuit regulates memory expression”, third paragraph and “A separate dopamine-glutamate circuit regulates memory consolidation”, first paragraph).

2) There are additional control experiments that are required.

a) In many figures (Figure 1, 2, Figure 1—figure supplement 4 and others), the variance appears to suggest the experiments are underpowered. Authors should show power analyses in the Materials and methods and make sure n is adequate.

b) Circuit comparisons for ethanol memory and sugar reward memory are done under different internal states (hungry vs. sated). It's been shown by many groups that the mushroom body centered circuits are engaged by cues depending on internal state. Moreover, the dopamine neuron for sugar reward – cue association is represented in MB299B – the PAM-α1. MB299B was tested, but the n is too low to be conclusive (compare shi/MB299B n=6 (Figure 1—figure supplement 4), versus shi/MB109B n=24 (Figure 2F)). Since the authors are making the comparison they should improve the data quality for MB299B, and demonstrate that sugar reward learning is defective in a parallel experiment, perhaps doing the ethanol and sugar tests under the same internal state.

c) Did the authors test whether flies choose equally between odor1 and odor1 + EtOH? Binary odorant mixtures are normally more attractive than the single odorant.

d) The authors' claim that ethanol enhances the calcium response in PAM neurons is not well substantiated (Figure 1E-F). Because the control measurement (odor only) is followed by the ethanol presentation, order-dependent effects are not excluded. Independent groups, one of which receives only odor and the other the odor and ethanol, should be compared to account for the enhancement

3) Authors need to temper some of their claims. First, although the authors have shown in the past that alcohol-trained flies overcame strong electric shocks during memory retrieval (Kaun et al., 2011), the current manuscript does not directly address the underlying mechanism so cannot claim they have shown how the alcohol-memory is transformed into an inflexible form. This has an impact on the title of this paper which is misleading. The data do not address whether memories are not changing. They only show that they might be more stable. In addition "forgetting" of learned information is an active process and Shuai et al., 2015 have shown that the dopaminergic PAM neuron are required for forgetting. It is not clear that the memory observed in this paper cannot be replaced another memory. Second, in Figure 5E the experimental is only different from one of the two controls, yet the authors interpret this as a significant result. In fact, it is the control that is low, the experimental is about the same magnitude as the non-significant experimentals in the other panels in the figure. The conclusions about the role of this neuron are not supported and should be removed from the manuscript. In general unless something is different from *all* the controls it should not be interpreted as significant.

4) The recent EM reconstruction of the alpha lobe shows that DANs all have DCVs. The manipulations used by the authors could block not just release of DA, but release of any co-transmitter as well. Authors should comment on this possibility.

---

## [Author Response]

Essential revisions:1) There are many problems with the way the paper is written. The references to the mammalian literature on dopamine function in Results are unnecessary and potentially misleading – there were no experiments done with mice or rats. The authors make comparisons or claims about many different areas of the addiction literature. If the authors wish to defend this breadth, then more accurate assessment of their experiments in relation to the areas of the literature is needed. For example, the prediction error literature in mammals is grounded in electrophysiological measurements and assessments of populations of neurons, whereas the experiments presented here are inactivation by dominant negative Shibire in genetically tagged neurons. Further, there is no information about whether the fly neurons are permissive or instructive for the behaviors, we do not know if they are firing action potentials, and if they are, we do not know if the firing is time locked to the cue, ethanol, or some aspect of the behavioral sequence. At minimum, the discussion of the scientific background and link to mammalian literature should all be moved to the Discussion. In addition, the text is unnecessarily long and can be more concise, especially for Figure 2, where single neurons have not actually been identified. Authors may also want to minimize repetitions, as this lengthens the paper unnecessarily (e.g. subsections “A dopamine-glutamate circuit regulates memory expression”, third paragraph and “A separate dopamine-glutamate circuit regulates memory consolidation”, first paragraph).

We appreciate the reviewer’s feedback and opinions on the writing of this manuscript and have substantially revised the text to accommodate their views. Although we originally included the mammalian literature to communicate the literature and rationale that inspired much of the experiments performed in this work, we have reviewed and removed the majority of the references to this literature in the Results section. We have also moved Figure 2 to Figure 1—figure supplement 5 and minimized the discussion of this data. We hope the reviewers feel that these edits sufficiently strengthen the manuscript.

2) There are additional control experiments that are required.a) In many figures (Figure 1, 2, Figure 1—figure supplement 4 and others), the variance appears to suggest the experiments are underpowered. Authors should show power analyses in the Materials and methods and make sure n is adequate.

Rather than performing a power analysis, prior to beginning our experiments we used our knowledge of typical sample size in past experiments to inform the most appropriate N for the experiments used in this study. Our rationale was to design the experiments such that the behavioral effect was replicated three separate times. As a consequence, our sample size is similar to other *Drosophila* behavioral experiments and generously exceeds what is typically used in rodent animal models. Although it is helpful to perform initial power calculations, our understanding is that it is not appropriate to perform post hoc power analyses as the observed power is a simply a function of the p-value. We respectfully refer the reviewer to Hoening and Heisey, 2001 publication “The abuse of power: The pervasive fallacy of power calculations for data analysis.” in the American Statistician 55(1):19-24. We did add more N to MB032B and MB058B retrieval so their numbers were comparable to other groups.

b) Circuit comparisons for ethanol memory and sugar reward memory are done under different internal states (hungry vs. sated). It's been shown by many groups that the mushroom body centered circuits are engaged by cues depending on internal state. Moreover, the dopamine neuron for sugar reward – cue association is represented in MB299B – the PAM-α1. MB299B was tested, but the n is too low to be conclusive (compare shi/MB299B n=6 (Figure 1—figure supplement 4), versus shi/MB109B n=24 (Figure 2F)). Since the authors are making the comparison they should improve the data quality for MB299B, and demonstrate that sugar reward learning is defective in a parallel experiment, perhaps doing the ethanol and sugar tests under the same internal state.

The reviewer brings up a valid point regarding the influence of different internal states. We originally addressed this in the Discussion, however, we have since removed the circuitry comparison from Figure 2. Despite the fact that alcohol reward memory is present in hungry and sated flies, it is not possible to test 24hr sugar reward memory in sated flies. It is an interesting idea to test alcohol reward memory in hungry flies to see if the circuits are similar, however, we feel this is out of the scope of this paper and would require redoing all behavior experiments. Because MB299B was part of a circuitry screen, and not the hypothesis-driven experiments in the main text, it had a lower N in the first submission of the manuscript. We have since increased the N for this driver in sated flies and added this data to the main figure (now Figure 2). MB299B takes the place of MB087C which has lower expression in the same PAM-β`2a neurons (level 1 according to FlyLight and Aso et al., 2014). However, even with the additional N, we still do not see a requirement of PAM-α1 in the acquisition or retrieval of alcohol reward memory. It is possible that the involvement of α1 is limited to internal states of hunger. We elaborate on this point in the Discussion (fourth paragraph).

c) Did the authors test whether flies choose equally between odor1 and odor1 + EtOH? Binary odorant mixtures are normally more attractive than the single odorant.

The rationale for this suggested experiment is unfortunately not clear to us. To clarify our current behavioral protocol, at test: flies were presented with two individual odors, the odor that was not previously paired with alcohol and the odor that was previously paired with alcohol. The pair odor presented at test was not mixed with EtOH; thus, preference for the paired cue 24 hours later cannot be explained by a binary odorant effect. In fact, in Kaun et al., 2011, we showed that the presentation of the paired odor and ethanol in the absence of intoxication was not enough to elicit preference, but instead the flies needed to experience the effects of intoxication in order to develop a preference for the paired odor 24 hours later, which rules out the influence of binary odor mixtures influencing behavioral preference. We would be happy to accommodate the reviewer if they could explain in more detail their expectations for their proposed changes/experiments. As it is, we aren’t sure why the proposed odor 1 vs. odor 1 + ethanol test would be relevant to the ideas currently presented in the manuscript. We apologize if the behavioral protocol was not obvious. We have clarified the description of the protocol in the text (subsection “Dopamine neurons innervating the mushroom body are required for alcohol reward associations”, second paragraph).

d) The authors' claim that ethanol enhances the calcium response in PAM neurons is not well substantiated (Figure 1E-F). Because the control measurement (odor only) is followed by the ethanol presentation, order-dependent effects are not excluded. Independent groups, one of which receives only odor and the other the odor and ethanol, should be compared to account for the enhancement

To address this point, we performed additional calcium imaging studies in which the two odors used in our memory assay were presented sequentially. We counterbalanced the order of odors across animals and show that while odor 2 tends to be higher than odor 1, it is higher in both Early and Late Epochs and the differences are not significant (see Figure 1—figure supplement 3). We don’t see evidence of responses to odor 2 being lower during the Early Epochs and higher during the Late Epochs as we reported in Figure 1. We have also performed calcium imaging experiments that measured dopamine response to the presentation of just ethanol. Interestingly, we don’t see the same increase that we reported in Figure 1. We hypothesize that the effects we are seeing are a result of the association between odor and ethanol and further hypothesize that these effects incrementally increase across multiple sessions. We have updated the text in the Results section to account for these additional control experiments (subsection “Dopaminergic encoding of alcohol memory acquisition occurs at the population level”, second paragraph). Since the unconditioned stimulus (ethanol) induces pharmacological properties that persist beyond stimulus offset, it is not possible to gain an understanding of how these memories are acquired using the opposite order, as is conventional in classical conditioning assays. However, we believe this effectively demonstrates that odor order is not what is driving the neuronal responses in Figure 1.

3) Authors need to temper some of their claims.

We appreciate the feedback and opinions of the reviewer and are happy to accommodate.

First, although the authors have shown in the past that alcohol-trained flies overcame strong electric shocks during memory retrieval (Kaun et al., 2011), the current manuscript does not directly address the underlying mechanism so cannot claim they have shown how the alcohol-memory is transformed into an inflexible form. This has an impact on the title of this paper which is misleading.

Although the title of the paper does not mention inflexibility, we have updated the title of the paper, to “circuits that encode and guide alcohol-associated preference” which we feel is a more accurate summary of the data.

The data do not address whether memories are not changing. They only show that they might be more stable. In addition "forgetting" of learned information is an active process and Shuai et al., 2015 have shown that the dopaminergic PAM neuron are required for forgetting. It is not clear that the memory observed in this paper cannot be replaced another memory.

We believe that the current manuscript provides a circuit framework for understanding the inflexible nature of alcohol-associated memories. We show that alcohol reward memories are encoded via a population of dopaminergic neurons involved in reward memory that progressively increase their activity as the flies become intoxicated. We suggest that this dynamic is what makes alcohol-associated memories stronger than adaptive memories such as sucrose, which as previously reported (Kaun et al., 2011) do not impel flies to overcome a shock of the same strength. It was never our intention to show that alcohol-associated memories can be replaced with another memory. In fact, the majority of research at present suggests that a memory is not replaced with another, but is either updated with new information (reconsolidation) or competes with a new memory (extinction). Even the elegant work of R. Davis and colleagues does not claim there is no molecular trace of the “forgotten” memory. We believe the inflexibility of pathologically enhanced memories such as AUD lies in the fact that despite an aversive consequence flies still show preference for cues associated with alcohol intoxication. Behavioral inflexibility in the face of aversive consequences is present in a number of psychiatric disorders and thus an important lens to study AUD and other disorders. However, because, as the reviewer suggests, we don’t directly test flexibility, we have removed the use of inflexible from the results and limited its use to the Introduction and Discussion. We hope the reviewers feel that these changes substantially strengthen the manuscript.

Second, in Figure 5E the experimental is only different from one of the two controls, yet the authors interpret this as a significant result. In fact, it is the control that is low, the experimental is about the same magnitude as the non-significant experimentals in the other panels in the figure. The conclusions about the role of this neuron are not supported and should be removed from the manuscript. In general unless something is different from all the controls it should not be interpreted as significant.

With regard to MB074C, it’s true that the controls have a lower preference. We hypothesize this is likely state-dependent as the flies have been in a 30C incubator nearly 24 hours. State dependent changes are a separate line of research in the lab. However, we agree it is problematic that the experimental is only significantly different from the UAS control. To address this, we have re-run these experiments with MB002B, which has similar expression levels in MBON-β`2mp, and gave flies longer to habituate prior to testing. Strikingly with MB002B we see a significant reduction in preference for cues associated with alcohol relative to both controls when D2-receptors are knocked down in this line and a significant enhancement when neurotransmission is silenced during consolidation. We have added this data to what is now Figure 4.

4) The recent EM reconstruction of the alpha lobe shows that DANs all have DCVs. The manipulations used by the authors could block not just release of DA, but release of any co-transmitter as well. Authors should comment on this possibility.

We fully acknowledge that dopamine neurons are heterogeneous and co-release transmitters (for example Aso et al. *eLife*, 2019). In order to show that the effects we were seeing were indeed due to dopamine, we performed dopamine receptor knockdown experiments in the post-synaptic neurons. These results suggest that dopamine receptors in MBON-β`2mp are required for alcohol-associated memory (see updated Figure 4). We have also performed dopamine auto receptor knock-down experiments in the population of PAM neurons and see a decrease in alcohol-associated memory which suggests that the regulation of dopamine release is also required. We have added this data to Figure 1. We recognize this does not rule out the influence of cotransmitter release and have thus added text to acknowledge this possibility (Discussion, second paragraph).